# Potency and Powder X-ray Diffraction (PXRD) Evaluation of Levothyroxine Sodium Tablets under Ambient, Accelerated, and Stressed Conditions

**DOI:** 10.3390/ph17010042

**Published:** 2023-12-27

**Authors:** Mercy A. Okezue, Stephen R. Byrn, Josiah Probost, Madison Lucas, Kari L. Clase

**Affiliations:** 1Department of Pharmaceutical Sciences, University of Michigan, Ann Arbor, MI 48109, USA; 2Industrial and Molecular Pharmaceutics Department, Purdue University, West Lafayette, IN 47907, USA; probst1@purdue.edu (J.P.); lucas146@purdue.edu (M.L.); 3Biotechnology Innovation and Regulatory Science (BIRS) Center, Department of Agricultural and Biological Engineering, Purdue University, West Lafayette, IN 47907, USA; kclase@purdue.edu

**Keywords:** FDA, enforcement report, stability, reference standards, degradation, levothyroxine

## Abstract

Levothyroxine tablets, although highly prescribed in the United States, have been one of the most frequently recalled products. Because of the importance of the medication, several efforts have been put in place by the United States Food and Drug Administration (US FDA) to control the quality of levothyroxine tablets available to patients using the drug. The choice of excipients used in the formulation has been shown to impact the hygroscopicity and microenvironment, and ultimately the stability of the levothyroxine tablets formulations. Based on information generated from the US FDA Enforcement Report database, one of the main reasons for recalls is the low potency of different batches of the product. The yearly product recall trends for levothyroxine formulations were determined using the FDA Enforcement Report database. Three brands of levothyroxine tablets were selected with excipient lists similar to those products that have been historically recalled. The samples were placed at ambient (~23 °C), accelerated stability (40 °C/75% RH), and stress (50 °C/75% RH) conditions for up to 6 months. Sample potencies were determined at 0, 1.5, 3, and 6 months using the methods for assay and impurities in the United States Pharmacopeia (USP) monograph for levothyroxine tablets. Additional sample monitoring was conducted by overlaying the initial powder X-ray diffractograms (PXRD) of the samples from 0 months with the patterns generated thereafter. There has been a decline in the number of levothyroxine tablets recalled over the years. The highest numbers of recalls were recorded in the years 2013 [33] and 2020 [23]; no recalls occurred in the years 2019 and 2022. All of the brands evaluated met the USP 95.0–105.0% assay requirements at 1.5 months under accelerated conditions; only one of the brands complied at 3 months. Under ambient conditions, two brands were stable at 6 months, with borderline assay results. For stability, levothyroxine was found in microgram quantities in the formulations and PXRD could not detect changes at these low levels. However, we found some distinguishing data for samples under stress conditions.

## 1. Introduction

Levothyroxine sodium tablets (L-T_4_) are listed among the 200 most prescribed drugs in the United States. Levothyroxine sodium is available in the form of tablets and capsules, and is sometimes compounded for administration as an oral solution. It is commonly used in patients with insufficient or absent thyroid hormones T_4_ (tetraiodothyronine or thyroxine) and T_3_ (triiodothyronine or Liothyronine), to treat cretinism, myxedema, nontoxic goiter, and hypothyroidism [1,2,3]. Despite the importance of this drug, regrettably, L-T_4_ tablets are one of the molecules that have been highly recalled by the FDA for issues related to sub potency of the active pharmaceutical ingredient (API) in the products [4]. There are concerns about the stability of formulated L-T_4_ tablets and ensuring product quality throughout the approved shelf life. To evaluate the amount of API available to patients over the expected expiry dates on the product labels, we investigated the X-ray diffractions of powdered samples and the chemical potencies of three brands of levothyroxine tablets throughout 6 months of normal, accelerated, and stressed conditions.

Thyroxine from thyroid hormones maintains essential biologic functions related to the reproduction, growth, and development of the brain, as well as other vital organs like the heart [5,6,7,8,9,10]. Levothyroxine (L-T_4_) was introduced as a United States Pharmacopeia (USP) compendial monograph in 1933 and is the synthetic form of the human endogenous hormone, thyroxine [11]. T_4_ is converted by 5′-deiodinases to its active metabolite T_3_ with a ~4× higher potency and exerts the majority of the physiological effects of thyroid hormones [12]. Although hypothyroidism has multiple etiologies and manifestations, patients with this disorder generally exhibit symptoms like fatigue, muscle cramps, voice changes, dry skin, poor tolerance to cold temperatures, and constipation [13]. L-T_4_ is administered at low doses (25–500 µg/day) and has a narrow therapeutic index [14,15]. The low dose: excipient ratios in levothyroxine preparations pose a risk for delivering satisfactory content uniformity in each of the dosage formulations. The chemical potencies of the samples of levothyroxine sodium tablets evaluated in this study were conducted using certified reference standards from USP.

Stability issues were recorded from the HPLC analyses of generic levothyroxine sodium pentahydrate (L-T_4_) tablets as early as the 1960s, after the first brand of sodium salt was marketed as Synthroid^®^ in 1955 [16,17]. The choice of excipients used in manufacturing tablets is a common source of variability in product performance. Although considered inert materials, the choice of excipients impacts the tablet performance, including the dissolution and stability [18,19]. One of the plausible causes of instability in L-T_4_ tablets was linked to excipients that impact the hygroscopicity and microenvironment of the formulations [4]. An earlier study suggested that levothyroxine API was stable at accelerated conditions for up to 6 months. Excipients such as lactose anhydrous, starch, or microcrystalline cellulose introduced a higher moisture content (≥5%) and a loss of API potency, while dibasic calcium phosphate or mannitol achieved a lower moisture content (<1%) and maintained the API potency for up to 3 months. The study also reported that adding basic pH modifiers (e.g., sodium carbonate, sodium bicarbonate, and magnesium oxide) improved the stability of L-T_4_ tablets for up to 6 months [20]. Other findings evaluated L-T_4_ as mono or binary component systems with other excipients. They characterized the samples with diverse spectroscopic and chromatographic instrumentation methods to decipher potential formulation failure modes [21,22]. Such studies provide enormous benefits to the pharmaceutical manufacturers of the product. However, our study evaluated the finished products containing several excipients with multi-faced possible interactions on the microenvironment of the API, thereby providing real-world data for the formulations.

The influence of the microenvironment on the stability of L-T_4_ tablets may be related to the aqueous solubility of levothyroxine sodium, which varies in pH at 25 °C. The solubility decreased from pH 1 to 3, was constant between 3 and 7, and increased at a more basic pH > ~7. The ionizable groups of thyroxine were the carboxyl group (pKa = 2.4), phenolic group (pKa = 6.87), and an amino group (pKa = 9.96), and depending on the pH of an aqueous media, T_4_ exists either as a cation, zwitterion, anion, or dianion species (see Figure 1) [20,23]. The stability of L-T_4_ formulations containing mannitol or dibasic calcium phosphate was improved by creating a more basic microenvironment with excipients that are pH modifiers (e.g., sodium bicarbonate and magnesium oxide). Conversely, the potency of the tablets declined with citric and tartaric acids [20].

Won described the kinetics of degradation for T_4_ and suggested that deiodination to T_3_ was the major pathway in aqueous media and deamination for the solid state. The study illustrated a decrease in the stability of levothyroxine sodium in aqueous media at a lower pH via first-order kinetics, and a biphasic behavior for solid-state degradation at higher temperatures [23]. This work expanded on the assertion from a team of FDA scientists that the instability of T_4_ was responsible for the low assay values experienced by pharmaceutical manufacturers of the product [16]. Early concerns about several cases of product failures from L-T_4_ instability, including within-lot discrepancies in tablet potencies, resulted in the FDA declaring unapproved oral forms of the drug as “not safe” [3]. However, medical necessity for the drug, in effect, required the agency to allow manufacturers who submitted new drug applications for the product by 14 August 2000 to continue to market the drug with a shelf life of up to 18 months. The FDA mandated manufacturers to stop the distribution of unapproved L-T_4_ tablets by 14 August 2003 [24], and recommended a stability indicating reverse phase HPLC method for quality control of the tablets [3,25,26]. In the current study, we selected some brands of L-T_4_ tablets with different excipients and determined the potencies using assay methods 1 and 2 in the monograph of the current United States Pharmacopeia (USP) for the product.

Historically, L-T_4_ tablet recalls have been attributed to a loss of potency before the end of the shelf life of the formulations [4,11,20]. The purpose of this study was to use historical data from the FDA Enforcement database to identify the L-T_4_ formulations that were recalled based on chemistry manufacturing and control (CMC) failures, conduct stability studies on products with similar formulations, and to decipher if the potencies complied with the USP limits for assays and impurities. In addition to the official test methods approved by the USP, we investigated the adequacy of an additional spectrometric technique.

During the early stages of drug formulation design, a combination of several spectroscopic and thermoanalytical methods, like hot-stage microscopy (HSM), differential scanning calorimetry (DSC), scanning electron microscopy (SEM), solid state nuclear magnetic resonance (sNMR), and powdered X-ray diffraction (PXRD), can be used to predict formulation incompatibility and avoid failures downstream [27]. Powder X-ray diffraction (PXRD) is a technique that can be used to determine the amorphous vs. crystalline nature of a material. Several studies have used PXRD for monitoring changes in APIs, especially when such transformations negatively impact the bioavailability of the drug. In one study, PXRD detected heat-induced amorphous transformation in the pharmaceutical testing of picotinamide with various pharmaceutical excipients [28]. A similar study used PXRD and other isothermal techniques to establish compatibility between propafenone HCl and lactose monohydrate [29]. Further studies reported PXRD as one of the spectroscopic methods for conducting API using excipient assessments to support the stability data for pharmaceutical products [30,31,32]. As there is evidence to support using PXRD to monitor formulation changes in pharmaceutical development, this study employed the technique as a tool to possibly detect changes in the diffractograms of the L-T_4_ formulations used in the stability studies. However, in the formulations studied, L-T_4_ was present in microgram quantities and the powder instrument used was not sensitive enough to detect changes at these low levels. Instead, changes in the PXRD represented changes in the excipients.

Stress tests on the finished pharmaceutical product were conducted to assess the effect of severe conditions on the drug product. A model was developed in the late 1990s to predict potential stability problems that may have arisen from API−excipient interactions in oral tablets by evaluating the potency after the product was stored at 50 °C for 1–3 weeks. These experiments were carried out at a more elevated temperature than recommended by the International Conference on Harmonization (ICH Q1A(R2)), namely 40 ± 2 °C/75% RH ± 5% RH for accelerated studies; the rationale behind this was to quickly identify excipients that would cause potential stability failures [33]. Thermal stress testing was used to evaluate the compatibility of ketorolac and empagliflozin with some common excipients, and it suggested the benefits of these studies for achieving a lower risk for formulation failure [30,34]. In addition to placing samples under the ICH recommended conditions for accelerated study, the current research investigated the potency of L-T_4_ tablets stored at 50 ± 2 °C/75% RH ± 5% RH as a potential predictor of the formulation’s accelerated and real-time stability.

## 2. Results

### 2.1. Recalls for Levothyroxine Formulation within the United States

The recall data for levothyroxine (L-T_4_) formulations are available from the FDA Enforcement Reports webpage (https://www.fda.gov/safety/recalls-market-withdrawals-safety-alerts/enforcement-reports (17 January 2023)). At the time of this report, 8 June 2012 was the earliest date available for all regulated products recalled within the United States. The data obtained were exported as comma-separated value (CSV) files and all analyses were conducted using formulas and graphics in Microsoft Excel^®^ 2019.

A total of 107 L-T_4_ formulations were recalled to from 2012 to 2022, with the highest numbers of recalls recorded in 2013 (33) and 2020 (23); no recalls occurred in 2019 and 2022. Approximately 71% of all L-T_4_ recalls were due to chemistry manufacturing control (CMC) failures; 28% were from packaging, container, and labelling issues; and ~1% were other factors. The sub-potency of the L-T_4_ formulations accounted for about 34% of the products recalled for CMC, while 57% was attributable to non-specified cGMP deviations. A summary of the amount of L-T_4_ formulations recalled from the US, as well as the reasons for the recalls, are provided in Figure 1. Using a risk-based approach, the study assessed the potency and impurity profiles of the study samples, as these parameters formed the highest percentage of CMC issues that could be verified in a quality-control lab.

### 2.2. Sample and Test Method Selection

Three brands of L-T_4_ tablets were purchased from the Purdue University Pharmacy. Two of the formulations were manufactured by firms that had FDA Enforcement reports for L-T_4_ tablets. Table 1 summarizes the details of these products, including the excipients in the formulations. All of the analytical tests, including the 6-month accelerated stability assessments, were conducted within the shelf life of these products. Based on the historical evidence that sub-potency is the main CMC reason for the recall of L-T_4_ formulations, the assay and impurity tests from the USP 2022 edition were used to evaluate the products in this study. Details of the USP reference standards and the relevant label instructions used for the tests are provided in Table 2.

### 2.3. Levothyroxine Products Assay

USP 2022 Edition has two assay methods in the monograph for L-T_4_ tablets (Limits 95.0–105.0%). The product leaflets for each formulation contained only information on the dissolution methods (except brand B); therefore, the two USP assay methods were used to assess the potency of each brand in this study. The assay results for the samples tested at 0 months and those placed under accelerated stability and stressed conditions are presented in Table 3. When interpreting the results, it was assumed that an assay result was satisfactory if the brand satisfied either USP method 1 or 2. All of the brands met the assay limits for L-T_4_ tablets at the beginning of the study, at 0 months. The stability samples placed at 40 °C/75% RH maintained their potencies after 1.5 months, but only brand C, stored under stress conditions of 50 °C/75%, was stable. By the third month, only brand A maintained its potency under the 40 °C/75% RH condition. All of the brands were sub-potent by the sixth month of accelerated stability conditions. The samples kept under ambient conditions (~23 °C) for 6 months had borderline assay results, and brand A was not within the USP limits. Unsatisfactory assay results were verified with replicate testing using fresh samples and reagents preparations.

### 2.4. Levothyroxine Products Impurities

At the end of the 6-month stability study, all of the brands of L-T_4_ evaluated met the USP requirements for impurities (acceptance criteria: NMT 2.0% of liothyronine sodium). Table 4 provides a summary of the impurity assessments for the formulations tested.

### 2.5. PXRD Tracking for the Levothyroxine Products

The intensities of the angular diffractions of the powdered L-T_4_ samples were measured between 4–40° 2θ. The PXRD of the samples measured at 0 months was used as a baseline to track possible changes that may have occurred in the formulation over time (Figure 2). Being generic products, the initial diffractograms were compared to the pattern deposited for the innovator product (Synthroid^®^) in the Cambridge crystallographic database center (CCDC) to observe for any match; but none was observed. The samples were then re-evaluated at 1.5 months, 3, and 6 months to identify possible structural changes detectable by PXRD. The fingerprints from the samples stored at 40 °C/75% RH did not show any significant difference at the end of the study. However, for samples kept under stress conditions of 50 °C/75% RH, additional diffraction patterns were observed for brands A and B at 3 and 6 months (Figure 3).

### 2.6. Verification of Equipment Used for Study

To ensure that all the major pieces of equipment used for this study were performing as expected and would yield the intended results, some levels of verification or calibration were conducted using traceable instruments. The protocols for these verifications were developed using USP and EDQM (European Directorate for Quality of Medicines) recommendations for qualifying pieces of equipment, and reports were generated.

The Agilent 1100 Series HPLC was verified for the following parameters: solvent delivery system flow rate (meet requirements for accuracy and precision), gradient composition accuracy and ripple (acceptance criteria for gradient composition: absolute deviation: ±2 of the adjusted value; acceptance criteria for ripple: ≤0.2%; comment: met both requirements), injector volume precision and carry-over (acceptance criteria: RSD of all peak areas obtained should be ≤1.0%, the results obtained 0.2%; acceptance criteria for carry-over: the percentage of the peak area corresponding to caffeine in the blank injection 2 is NMT 0.2%, the results obtained 0.09%), autosampler thermostat accuracy and precision (acceptance criteria: the actual temperature may not differ more than ±0.5 °C with respect to the selected temperature, the results obtained temperatures measured at 4.0 (°C), 8.0 (°C), 25.0 (°C), 30.0 (°C), and 40.0 (°C), which met the requirement), thermostarting accuracy (the temperature was set at 40 °C, with acceptance criteria of 38–42 °C, and the results obtained between 39.6 °C and 40 °C), and UV/DAD detector linearity (acceptance criteria: r^2^ ≥ 0.999, the results obtained r^2^ = 0.9999).For verification of the Melter Toledo Analytical weighing balance, the following parameters were performed using a box of standard weights: repeatability error, Er, using 50 mg and 500 mg weights (acceptance criteria: Er shall not exceed the MPE of the balance Er ≤ MPE balance, the results obtained a balance reading that met requirements for both weights); error of indication (*Ei*) was performed using 2 mg, 5 mg, 20 mg, 50 mg, 100 mg, 200 mg, and 500 mg weights with one replicate per weight, in increasing the loading mode (acceptance criteria: *Ei* ≤ MPEbalance/3; the results obtained met the requirements for all of the weights used); an eccentricity test was performed by reading the values after placing a 50 mg weight on the center of the pan and on the center of the four quadrants (acceptance criteria NMT 0.05; the results obtained for the weight readings for all points of the balance met the requirements); and accuracy was determined by testing if 2 mg, 5 mg, 10 mg, 20 mg, 50 mg 100 mg, and 500 mg were within 0.10% of the test weight value (the results met the requirements for all of the weights used). Details for the weights used: Manufacturer: FISHER, supplied by TROEMNER INC 6825 Greenway Avenue. PA 19142*215/724-0800, Serial # 14275, Tolerance Class S.The hot air ovens used were the Fisher Scientific Isotemp oven, ID # 1002906, set at 40 °C, and a BINDER mechanical oven set at 50 °C. For mapping the oven temperatures, we used the following: Thermometer ID: Traceable^®^ 437190205-05, S/N: 210719592, Cal due date: 19 August 2023. Acceptance Criteria: the temperatures obtained at each mapped area needed to be within ±2 °C of the required temperature conditions. The results obtained from mapping different parts of the oven are shown in Figure 4, indicating that all points mapped within the ovens were within the specified range for both ovens.To prepare the HPLC buffers, if required, the SympHony VWR SB301 pH meter, made in U.S.A, was calibrated as needed. An example of one of the calibrations carried out on the pH meter before use is illustrated in Table 5. Two buffers (manufacturer: METTLER TOLEDO, InLab^®^ Solution, Sachets 30 × 20 mL, 4.01 (lot 1F100C) and 7.00 (Lot 1F128B), made in Greifensee, Switzerland), were used for calibration of the pH meter.

## 3. Discussion

The US FDA works to ensure that the quality of medicines used within the country are appropriate for their intended use. The low potency of API is one of the main reasons behind the high level of recalls for levothyroxine (L-T_4_) tablets in the United States. From the trends obtained from the FDA Enforcement Report database, the high number of recalls for L-T_4_ formulations in 2013 (33) and 2020 (23) decreased in 2021 (4) and 2022 (0). This may be an indication that manufacturers are making more stable L-T_4_ formulations. The product lists for the samples tested in this study contained blends of some excipients that have been identified as stabilizers in the formulation. Product stability is enhanced by a better understanding of the influence of the microenvironment on an API. Sodium carbonate, sodium bicarbonate, and magnesium oxide, which serve as basic pH modifiers, have been identified as better substitutes over starch or microcrystalline cellulose [20]. Furthermore, an instrumental analysis of binary systems of excipients and levothyroxine sodium pentahydrate, suggested avoiding the use of lactose, mannitol, and sorbitol [22]. Therefore, research on the impact of excipients on the microenvironment of L-T_4_ may provide better guidance to manufacturers for stabilizing the formulations. This may also be a contributory factor to the downward trend in US FDA recalls for L-T_4_ tablets observed in more recent years.

### 3.1. Possible Role of Excipients on Levothyroxine Tablets Stability

This study investigated three brands of L-T_4_ formulations with different excipient lists. All of the brands maintained their potencies after 1.5 months of accelerated conditions (40 ± 2 °C/75% RH ± 5% RH). However, failures were documented for two brands (B and C) after 3 months of stability. By the 6th month, all brands were below the USP assay limits for L-T_4_ tablets. For the samples kept in their original containers with desiccants, under ambient normal room temperature of ~23 °C, at 6 months, the potency of brand A was below the allowable limits, while B and C were just at the borderline (95.3% and 95.6%, respectively) of the lower limit for the USP assay (95.0%). For the samples stored under stressed conditions (50 ± 2 °C/75% RH ± 5% RH), only brand C was stable after 1.5 months, while others were below the USP assay limits.

The role of excipients in the stability of pharmaceutical dosage forms has been highly documented and some excipients have been suggested to lower the pH of the microenvironment, which is detrimental to L-T_4_ formulations. A study identified lactose anhydrous, starch, and microcrystalline cellulose as potential excipients that can cause increasing instability in L-T_4_ formulations, and sodium carbonate, sodium bicarbonate, and magnesium oxide as better substitutes [20]. A more recent study that investigated L-T_4_ formulations with anhydrous lactose, microcrystalline cellulose, and starch reported product failures after 3 months under 40 °C/75% RH [4]. In this study, under accelerated conditions, only brand A, which contained starch and sodium bicarbonate as part of its excipients, was stable after 3 months. This finding should have supported the research that had suggested that sodium bicarbonate stabilizes levothyroxine formulations that contain starch [20]. However, for the samples kept under ambient conditions, brand A was also the only brand that failed to meet USP assay limits. Therefore, the study results could not establish if the addition of sodium bicarbonate and other excipients was able to eliminate the deleterious impact of starch as a “suspect” excipient in the formulation. Overall, we submit that there may be multiple factors, in addition to the microenvironment created by excipients, that determine the stability of levothyroxine tablets. The use of accelerated predictive studies (APS) can also be an additional approach to obtaining stability results in a timelier manner to determine excipient compatibility for levothyroxine formulations. Using APS, manufacturers of the product can carry out pre-formulation investigations using protocols that span over a 3–4-week period and combining extreme temperatures and RH conditions (40–90 °C)/10–90% RH [35].

### 3.2. Letothyroxine Tablet Stability under Real-World-Use Conditions

The three brands of levothyroxine tablets used in this study were all within their shelf-life expiry at the 6-month assessments. The samples for accelerated and stress conditions were transferred to other containers for the period of the study, but the ambient samples were capped and left in their original containers. The perturbing issue with the assay results was the failure of brand A and the borderline results for brands B and C after 6 months of leaving the samples at room temperatures. With these observations, we proposed conducting a follow-up study to evaluate the potencies of the same products under real-world conditions indicated for storage for patient use. We recognize a limitation in the current study where some samples for accelerated conditions were stored in other packaging. The proposed study accessed samples stored in the manufacturer’s original containers and product desiccants in order to better reflect real-world patient use scenarios.

### 3.3. Products’ Impurity Profiles

The levels of impurities for all of the brands of the L-T_4_ tablets evaluated in this study maintained acceptable limits of liothyronine, as specified in the USP monograph for the product. The brands’ levels of liothyronine were below 2.0% for up to 6 months for the ambient, accelerated, and stress conditions of storage. Although there was a record of possible penicillin cross-contamination as the reason for the recall of an L-T_4_ formulation in 2015, there were no records of product recalls from high levels of impurities in the data obtained from the US FDA Enforcement database. This suggests that the formulations were stabilized against de-iodination, and the conversion of levothyroxine to liothyronine was not an important risk factor for the formulations.

### 3.4. Powder X-ray Diffraction Patterns as a Tool for Monitoring Stability

PXRD was used as a technique to monitor the excipient compatibility of API [27,31,32]. The initial PXRD fingerprints obtained from the study samples were used as a baseline to monitor if changes in the formulation would be detectable through changes in their diffractograms. An earlier study evaluated synchrotron XRD patterns from a 200 mg tablet compressed from pure levothyroxine sodium pentaphosphate API. The tablet was transferred between the sample chambers maintained at 40 °C/0% RH and ~25 °C/75% RH, and the XRD patterns were collected over 3 h. At 40 °C/0% RH, they observed changes in the lattice structure, as well as ∼8% weight loss, leading to the formation of levothyroxine monohydrate. There was a reversal in the crystalline structure back to the pentahydrate form when the tablet was transferred to the second chamber, supporting the theory of dehydration as one of the mechanisms for L-T_4_ degradation [36]. The levothyroxine sodium evaluated in our study contained the API and various excipients with crystalline structures, which presented a challenge in monitoring the diffraction peaks attributable to only the pure drug. Furthermore, the results of all of the samples under accelerated studies (40 °C/75% RH) did not suggest PXRD as a valuable tool for monitoring product stability, probably because of the low ratio of API excipients in the formulations. However, under stress conditions (50 °C/75% RH), there were some differences in the PXRD overlay from the third month. This presents the possibility of monitoring a product’s stability with synchrotron, which has a higher sensitivity for evaluating X-ray diffraction.

In summary, in the formulations studied, L-T_4_ was present in microgram quantities and the powder instrument used was not sensitive enough to detect changes at these low levels. Instead, changes in the XRPD suggest changes in the excipients’ crystallinity patterns.

### 3.5. Reliability of the Study Data

This study was conducted at a university, where the laboratory conditions were not necessarily cGMP compliant. However, to ensure the reliability of the test results, all of the major pieces of equipment used in this study were adequately qualified/verified to assure the accuracy of the test results. The out-of-specification assay results obtained at 3 months were confirmed by performing a duplicate analysis with freshly prepared samples and reagents. The tests were performed by qualified personnel who have headed an analytical chemistry laboratory for ISO/IEC 17025 accredited for 17 test scopes [37].

## 4. Materials and Methods

### 4.1. Using Real-World-Data for Sample Selection

The US FDA Enforcement Report (FER) database (https://www.accessdata.fda.gov/scripts/ires/index.cfm#tabNav_advancedSearch (17 January 2023)) provides details of all regulated products that have been recalled by the agency. Under the advanced search feature of the website, a query using “levothyroxine” as the product description returned a list of all of the dosage forms containing the molecule that had been recalled within the mid-year of 2012 to 2022. From the database, details of the recalling firms, the NDC code information, and the reasons for recall enabled us to identify the brands of L-T_4_ tablets sampled for the study. The NDC codes of the levothyroxine brands were searched on the NIH Dailymed database (https://dailymed.nlm.nih.gov/dailymed/ (19 January 2023)) to obtain the excipient lists for the formulations. A simple convenient sampling of brands that had been recalled and a reference list of the products were selected for the study. Furthermore, based on literature evidence that certain excipients may contribute to stability failures in L-T_4_ tablets, two brands containing these “suspect materials” were selected and purchased from the pharmacy within the university. An additional brand was used as a control because it did not have the susceptible excipients in its product list.

### 4.2. Storage Conditions for Tests Samples

The study samples were stored under accelerated, ambient, and stressed temperatures and relative humilities (RH). Sufficient amounts of all products for the assay and impurity tests were placed under ICH Q1 recommended general storage conditions for accelerated stability studies of 40 °C/75% RH and real-time ambient conditions of ~23 °C. Additional samples were also placed at 50 °C/75% RH to explore L-T_4_ tablet properties at exacerbated conditions. The samples for ambient conditions remained capped in their original plastic containers and enclosed desiccant packaging materials, while the ones for accelerated studies were placed in opened containers. All of the samples for accelerated studies were stored inside tightly sealed desiccators containing a saturated brine solution which provided the required relative humidity (75%) and they were kept in ovens set at the appropriate temperatures. Prior to use, the ovens were mapped with traceable thermometers to verify they read ±5 °C of the set temperatures. The samples were withdrawn at intervals of 0, 1.5, 3, and 6 months, and United States Pharmacopiea (USP) tests were conducted to determine the stability status.

### 4.3. Potency Determinations by USP Tests

To carry out tests for assays and impurities, as per the requirements in the USP monograph for L-T_4_ tablets, current lots of certified reference standards (RS) for levothyroxine and liothyronine were purchased from the USP convention. The potency for all of the samples was conducted using the two methods for HPLC assay listed in the USP monograph for levothyroxine tablets; the levels of impurities generated under stability conditions were also assessed. The label information for both RS were applied in the calculations for the assay and impurity determinations.

#### 4.3.1. Levothyroxine Sodium Tablets USP Assay Methods 1 and 2

Briefly, the USP monograph states “use sample solution 2 for tablets labeled to meet the requirements of Dissolution Test 3”. For all other products, we used the sample solution. Mobile phase: Acetonitrile and water (4:6) containing 0.5 mL of phosphoric acid per liter of mixture. Solution A: Dissolve 400 mg of sodium hydroxide in 500 mL of water. Cool and add 500 mL of methanol. Diluent: Methanol and water (6:4) containing 0.5 mL of phosphoric acid per liter of mixture. Levothyroxine stock solution: 0.4 mg/mL of USP levothyroxine RS in Solution A. Liothyronine stock solution: 0.4 mg/mL of USP liothyronine RS in Solution A. Make a 1:100 dilution of this solution using the mobile phase. Standard solution: 10 µg/mL of levothyroxine from levothyroxine stock solution and 0.2 µg/mL of liothyronine from liothyronine stock solution in the mobile phase. Sample solution: transfer an equivalent to about 100 µg of levothyroxine sodium, from finely powdered tablets (NLT 20), to a centrifuge tube, add two glass beads, pipet 10 mL of mobile phase into the tube, and mix on a vortex mixer for 3 min. Centrifuge to obtain a clear supernatant, filtering if necessary.

Sample Solution 2 (for tablets labeled to meet the requirements of Dissolution Test 3): Place the appropriate number of tablets (see Table 6) into a suitable container, add 100.0 mL of diluent, and shake by mechanical means for at least 30 min, or until the tablets are fully disintegrated. Pass through a PTFE filter of 0.45-µm pore size.

Chromatographic system: mode: LC; detector: UV 225 nm; column: 4.6 mm × 25 cm; packing L10; flow rate: 1.5 mL/min; injection volume: 100 µL.

System suitability-sample: standard solution. Suitability requirements—Resolution: ≥5.0 between liothyronine and levothyroxine; RSD: ≤2.0% for the levothyroxine peak.

Calculate the percentage of the labeled amount of levothyroxine sodium (C_15_H_10_I_4_NNaO_4_) in the portion of tablets taken using Equation (1)
(*r_U_*/*r_S_*) × (*C_S_*/*C_U_*) × (*M_r_*_1_/*M_r_*_2_) × 100(1)

*r_U_* = peak response from the sample solution.

*r_S_* = peak response from the standard solution.

*C_S_* = concentration of USP levothyroxine RS in the standard solution (µg/mL).

*C_U_* = nominal concentration of levothyroxine sodium in the sample solution (µg/mL).

*M_r_*_1_ = molecular weight of levothyroxine sodium, 798.85.

*M_r_*_2_ = molecular weight of levothyroxine, 776.87.

Acceptance criteria: 95.0–105.0%.

#### 4.3.2. Levothyroxine Sodium Tablet USP Limit of Impurities

For the limit of liothyronine sodium, the USP monograph states “use sample solution 2 for L-T_4_ tablets labeled to meet the requirements of Dissolution Test 3”. For all other products, we used the sample solution. Mobile phase, liothyronine stock solution, standard solution, sample solution, chromatographic system, and system suitability: proceed as directed in the assay. Liothyronine standard solution: 0.2 µg/mL of liothyronine from liothyronine stock solution, in mobile phase. Acceptance criteria: ≤2.0% of liothyronine sodium”.

Calculate the percentage of liothyronine sodium (C_15_H_11_I_3_NNaO_4_) in the portion of tablets taken using Equation (2).
(*r_U_*/*r_S_*) × (*C_S_*/*C_U_*) × (*M_r_*_1_/*M_r_*_2_) × 100(2)

*r_U_* = peak response from the sample solution.

*r_S_* = peak response from liothyronine from the liothyronine standard solution.

*C_S_* = concentration of USP liothyronine RS in the liothyronine standard solution (µg/mL).

*C_U_* = nominal concentration of levothyroxine sodium in the sample solution (µg/mL)

*M_r_*_1_ = molecular weight of liothyronine sodium, 672.96.

*M_r_*_2_ = molecular weight of liothyronine, 650.98.

Acceptance criteria: ≤2.0% of liothyronine sodium.

### 4.4. Chromatographic Analysis for Study Samples

The HPLC used in this study was an Agilent Technologies 1260 Infinity II, 1200 Series. The equipment was fitted with an Agilent Eclipse XDB-CN 5µm 4.6 mm × 250 mm column, set to deliver 100 µL at 1.5 mL/min with a UV detector set at 225 nm. All of the reagents used were ACS grade and the acetonitrile (Fisher Chemical, Waltham, MA, USA) was HPLC grade. The mobile phase used was acetonitrile and water (4:6) containing 0.5 mL of phosphoric acid per liter of mixture. Solution A and the diluent were prepared as prescribed in the USP monograph and were used to prepare the RS and samples solutions for the assay and impurity tests. The final dilutions for the RS solutions were prepared to contain approximately 10 µg/mL of levothyroxine and 0.2 µg/mL of liothyronine in mobile phase. The two sample solutions prescribed in USP were prepared for each brand of the levothyroxine tablets studied. The first solutions were prepared to achieve a final concentration of ~10 µg/mL, while the second solutions contained ~20 µg/mL. For impurity assessments, the standard solution used for the assay method was replaced with 0.2 µg/mL of liothyronine to determine the limits of liothyronine present in the levothyroxine tablets.

### 4.5. Powder X-ray Diffraction

To investigate whether possible changes in L-T_4_ tablets could be monitored using their powder X-ray (PXRD) patterns, diffractograms of the samples were collected at 0 months and served as the baseline for comparing the other PXRD collected at future times. The PXRD data were collected using a PANalytical Empyrean X-ray diffractometer equipped with Bragg–Brentano HD optics, a sealed tube copper X-ray source (λ = 1.54178 Å), fitted with a PixCel3D Medipix detector. All of the tablet samples were pulverized using an agate mortar and pestle and packed into a silicon single-crystal zero-background sample holder, 16 mm wide and 0.25 mm deep. The data were collected between 4° and 40° in 2θ under ambient conditions.

### 4.6. Qaulity Control Measures to Ensure Validity of Study Data

All of the major pieces of equipment, including the HPLC, analytical weighing balance, and pH meter, used in this study were adequately qualified or verified with traceable instruments, to assure the accuracy of the obtained tests results. Calibrated class A glassware and micropipettes were used for quantitative transfers and volumetric analysis. Furthermore, in line with the best international laboratory practices, all of the specification (OOS) results were thoroughly investigated for possible lab-related causes of error in the sample testing. When an OOS result was not traceable to a lab mistake, duplicate analyses using freshly prepared solutions and materials were conducted to confirm the assay values obtained from the USP methods. USP reference standards were used for the analysis, and the correction factors indicated on the label information were applied to the calculations in the USP L-T_4_ tablet monograph for both the assay and impurities.

## 5. Conclusions

There has been a decrease in the number of recalls of levothyroxine formulations in recent years. The potencies for three brands of levothyroxine tablets were assessed under ambient, accelerated, and stress conditions. The assay results indicate that the products were stable at 1.5 months under accelerated conditions of 40 °C/75% RH, but in the third month, only one brand maintained its potency. Only one of the brands was stable under stressed conditions, 50 °C/75% RH at 1.5 months. At 6 months, under an ambient storage temperature of ~23 °C, one out of the three brands did not meet the USP limit for assay (95.0–105.0%), while the potencies of the other brands were at the borderline of the lower limits. PXRD was not useful for monitoring the stability of the formulation. The three formulations were found to have acceptable limits of liothyronine impurity under all conditions of storage for up to 6 months.

## Data Availability

The datasets used and/or analyzed during the current study are available from the corresponding author upon reasonable request.

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
