# Peer review of "Potency and Powder X-ray Diffraction (PXRD) Evaluation of Levothyroxine Sodium Tablets under Ambient, Accelerated, and Stressed Conditions"

_pharmaceuticals, 2023, doi:10.3390/ph17010042_

Round 1

Reviewer 1 Report

Comments and Suggestions for Authors

Overall, the work presente din teresting for the audience fo the journal, however, several corrections shoudl be addressed before publication. 

1. The introduciton is not subdivided in sections. Please amend. 

2. However, the methodology shoudl be subdivided in section to make it clear for the reader. Much more detailed shoudl be indluded in each section to actually being able to reproduct the results if neccesary. 

3. How these bacthed were selected?

4. Statsitical analysis shoudl be included when neccesary. 

5. Apart from stability, does the dissolution profile gets altered after exposition to different temepratures adn relative humidities?

6. Which is the activation energy and humidity sensitivyt factor for the drug? I think this value shoud leb calculated. Check paper: Drug stability: ich versus accelerated predictive stability studies

7. The discussion is very poor. I think you shoudl describe the reperscusion adn other work related to accelerated stability studies and how these coudl be implemented. See papers: "Application of Accelerated Predictive Stability Studies in Extemporaneously Compounded Formulations of Chlorhexidine to Assess the Shelf Life" and "Guiding Clinical Prescription of Topical Extemporaneous Formulations of Sodium Cromoglycate Based on Pharmaceutical Performance"

Comments on the Quality of English Language

English needs revision. 

Author Response

Dear Reviewer,

Thank you for taking your time to provide feedback on our manuscript. Please find our responses to all the comments you provided in the document attached.

Reviewer 2 Report

Comments and Suggestions for Authors

The work submitted for evaluation concerns the assessment of the quality of the finished medicinal product. The problem is related to the recalled from the market a series of drugs. The authors present the methodology for assessing drug stability, and the possibility of check out the product quality using the PXRD technique. The fundamental question: what is the point of such tests on a product that has been recalled? What is the novelty of this work? If the described studies were to confirm the drug's instability, such a decision had already been made when it was withdrawn from the market. However, if they were to indicate possible reasons why this is happening, comprehensive and parallel tests should be carried out on the following samples (under the same stress conditions): 1- single API, 2- single excipients, 3-mix of excipients present in a given drug form, 4-mix API + single excipients, 5-mix API + mix of excipients present in a given drug form (= theoretical composition of a drug), 6-tablets made of API + excipients present in a given drug form, 7- the same as in 6 but placed in appropriate packaging (as model drug), 8 ready-made drugs (from the market). Moreover, what is the justification for presenting PXRD as a technique that allows for monitoring drug formulations? Especially when the authors proved that this technique is not useful in such cases? And they don't offer anything new in return?

Minor comments:

Abbreviations should not be used in the title of the work; rather full names.

There are no addresses in the authors' affiliations.

Abbreviations should be explained where they appear first in the text (e.g. HSM, DSC, etc.).

The descriptions in the Figures are illegible (Fig.1,3).

Author Response

(The authors gave the same response as above.)

Reviewer 3 Report

Comments and Suggestions for Authors

The topic is interesting and the paper is very well organized. It discusses in-depth details about stability of Levothyroxine Sodium tablets using XRD tool.  This paper worth consideration in my opinion after many revisions. 

1-     Full name for PXRD should be added for the title

2-     In the abstract, full name for all abbreviations should be stated when first mentioning e.g. US FDA; API in line 48. HSM, DSC, SEM, solid state NMR in line 122. ICH in line 143; CMC in line 160.

3-     In the abstract, please added the paragraph in lines 136-137;

[in the formulations studied, L-T4 is present in microgram quantities and the

powder instrument used was not sensitive enough to detect changes at these low levels.

Instead, changes in the XRPD represents changes in the excipients.]

This is very important note for readers.

4-     For keywords, add Levothyroxine. FDA could be removed

5-     Resolution of scheme 1 should be improved

6-     The USP Methods 1 and 2 (M1, M2) details should be stated either in the main text before table 3 or in supplementary file

7-     In the discussion, the authors should suggest more reliable and informative tool than PXRD for evaluation of L-T4 and its impurities e.g. mass spectrophotometry, IR, and NMR . Combination of more than one tool will be very efficient for assay of accelerated drug stability.

8-     Materials and methods should be subtitled.

9-     Study limitation and future plan should be provided.

10-  Comparison of the results with MS results is strongly recommended.  Identification of impurities should be illustrated using reliable tools.

11-  Comparison of efficiency of PXRD tool with universal attenuated total reflectance- Fourier transform infrared spectroscopy UATR-FTIR spectroscopy and thermal analysis is strongly recommended for the evaluation of  L-T4 stability

https://doi.org/10.3390/pharmaceutics12010058

12-  The merits and novelty over the following papers should be highlighted in the introduction.

LedeÈ›i, I., Romanescu, M., Cîrcioban, D., LedeÈ›i, A., Vlase, G., Vlase, T., Suciu, O., Murariu, M., Olariu, S., Matusz, P. and Buda, V., 2020. Stability and compatibility studies of levothyroxine sodium in solid binary systems—instrumental screening. Pharmaceutics12(1), p.58.

Shah, H.S., Chaturvedi, K., Hamad, M., Bates, S., Hussain, A. and Morris, K., 2019. New insights on solid-state changes in the levothyroxine sodium pentahydrate during dehydration and its relationship to chemical instability. AAPS PharmSciTech20, pp.1-10.

Best wishes  

Author Response

(The authors gave the same response as above.)

Round 2

Reviewer 1 Report

Comments and Suggestions for Authors

Authors have addressed correctly the comments so the paper is ready for submission.

Comments on the Quality of English Language

Ok

Reviewer 2 Report

Comments and Suggestions for Authors

-

Reviewer 3 Report

Comments and Suggestions for Authors

the authors did most of required changes. the paper could be published in my opinion 

best wishes